# Assessing Forest Type and Tree Species Classification Using Sentinel-1 C-Band SAR Data in Southern Sweden

**Alberto Udali** [1],*, **Emanuele Lingua** [1] and **Henrik J. Persson** [2]

1   Department of Land, Environment, Agriculture and Forestry, University of Padua—Viale dell'Università 16, 35020 Legnaro (PD), Italy; emanuele.lingua@unipd.it
2   Department of Forest Resource Management, Swedish University of Agricultural Sciences (SLU), 901 83 Umeå, Sweden; henrik.persson@slu.se
*   Correspondence: alberto.udali@unipd.it

**Abstract:** The multitemporal acquisition of images from the Sentinel-1 satellites allows continuous monitoring of a forest. This study focuses on the use of multitemporal C-band synthetic aperture radar (SAR) data to assess the results for forest type (FTY), between coniferous and deciduous forest, and tree species (SPP) classification. We also investigated the temporal stability through the use of backscatter from multiple seasons and years of acquisition. SAR acquisitions were pre-processed, histogram-matched, smoothed, and temperature-corrected. The normalized average backscatter was extracted for interpreted plots and used to train Random Forest models. The classification results were then validated with field plots. A principal component analysis was tested to reduce the dimensionality of the explanatory variables, which generally improved the results. Overall, the FTY classifications were promising, with higher accuracies (OA of 0.94 and K = 0.86) than the SPP classification (OA of 0.66 and K = 0.54). The use of merely winter images (OA = 0.89) reached, on average, results that were almost as good as those using of images from the entire year. The use of images from a single winter season reached a similar result (OA = 0.87). We conclude that multiple Sentinel-1 images acquired in winter conditions are feasible to classify forest types in a hemi-boreal Swedish forest.

**Keywords:** SAR; backscatter; forest classification; C-band; Sentinel-1





## 1. Introduction

Accurate and complete information about forests is required to fully understand the carbon balance and forest cover changes over time. Satellites are suitable for supporting this by acquiring images frequently and globally [1]. At a global level, the main characteristics to know about forests are their status and extent [2]. In order to keep this kind of information updated, the role of satellite images is beneficial for monitoring land use and its changes [3]. At a local level (regional or property) more detailed information is necessary to assess the status of a forest. Some information is provided by National Forest Inventories (NFIs) and, at property level, by local field plots implemented in management plans. The forest species composition is important when assessing, e.g., forest biodiversity and ecosystems considering also functions and processes, structures and services [4]. Knowledge about forest species composition can support the establishing of management plans to reach different goals, e.g., game and hunting, maximizing volume production, or enriching biodiversity and conservation [5]. As reported by the Forest Resources Assessment in 2020 [2], public interest is focused on forest change, e.g., degradation and deforestation, although many forests have changed in other ways. These other changes may not be immediately visible to the general public, such as in composition and density. Climate change is expected to cause substantial shifts in tree species distribution and forest structure [6]. Such impacts stress the need for continuous monitoring of forest changes that are predicted to be more rapid in the future [7].

The use of remote sensing (RS) techniques in forestry has increased rapidly during the last decades, to obtain information about forests at large scales (e.g., the boreal forest). RS tools can provide updated information to estimate quantitative variables such as tree height, volume and diameter. Furthermore, RS can be used to estimate qualitative data such as tree species, biodiversity [4], habitat loss and degradation, and the spread of invasive species [8]. RS images have long been used in forestry, earlier mainly relying on aerial photographs [9], but more data are now becoming available from other sources. Airborne platforms are still frequently used to provide ecological parameters at high resolution, e.g., leaf area index, species composition, canopy cover and gap closure [10]. In a review paper, Yu et al. [11] provided a solid comparison between different RS sources when used to retrieve forest variables (e.g., height, volume and diameter) at plot level. Fassnacht et al. [12] noticed that the number of tree species classification studies using RS has increased over the past 40 years, with emphasis on laser data and optical multispectral systems. Radar sensors have been used less, although these can provide data at intermediate resolution (from meters to tens of meters) and as long time series [11]. Radar data have been used in research related to forestry at least since the 1990s [2,13]. The Copernicus Earth Observation program by the European Space Agency (ESA) has facilitated the use of continuous synthetic aperture radar (SAR) data through the Sentinel satellites, starting in 2014. They guarantee a continuous data flow of images suitable for forestry applications. SAR can be used to create two- and three-dimensional reconstructions covering large areas [14]. The information provided by SAR (e.g., radar backscatter) is related to the Earth's surface, trees and vegetation canopy [12,15]. The wavelength of a SAR sensor determines the penetration of the transmitted microwaves [15,16]: longer wavelengths (e.g., L- and P-band) penetrate deeper into the vegetation canopies, whereas shorter wavelengths (e.g., C- and X-band) are reflected by small objects such as leaves and branches. SAR data are able to retrieve information about the forest continuously, to monitor surface cover and forest biomass [17], and to detect changes in the forest status [18], e.g., deforestation and land degradation [19], and the detection of natural hazards [20,21]. Many SAR studies have focused on the discrimination of land types [22,23], and fewer have addressed tree species classification, although SAR images with different wavelengths can sometimes be used complementarily [24].

Recently, various studies have used short wave SAR at C-band to investigate the interactions between the backscatter and trees' leaves, crown structure and canopy structure [25]. Dostálová et al. [26] explored how the signal reflection was affected by different species, branch geometry, and canopy structure. Frison et al. [27] demonstrated that there is a correlation between the radar backscatter coefficient of different polarizations and seasonality, and this is especially related to the phenology of the different species. Furthermore, Rüetschi et al. [28] reported an opposite behavior between coniferous and deciduous trees regarding annual monitoring of the backscatter in the winter and summer, i.e., leaf-on and leaf-off conditions. When performing classification, grouping of species into forest types delivered higher accuracy results compared to single tree species. Dostálová et al. [29] investigated the use of smoothed backscatter time series to perform forest type classification in various climatic regions and biomes in central and northern Europe. Rüetschi et al. [28] analyzed the application of a time series of dual-polarization acquisition over mixed forests in northern Switzerland to perform both forest type and species classification. Recent studies have shown the potential of using Sentinel-1 data for forest type classification [28–30], but they pointed out a need for improvements for boreal and hemi-boreal forests.

The objective of this paper is to evaluate the accuracy of area-based forest type and tree species classification using C-band SAR backscatter. We also want to investigate how the temporal stability of the backscatter may affect the prediction of tree species. Can the forest classification be improved with the use of acquisitions from multiple seasons and multiple years?

## 2. Materials and Methods

### 2.1. Study Areas

The study area is located in the Remningstorp forest estate and the nearby Eahagen natural reserve in southern Sweden (58°30′N, 13°40′E) (Figure 1). According to the management plan of 2008, the prevailing tree species in the area are Norway spruce (*Picea abies* (L.) Karst.), Scots pine (*Pinus sylvestris* L.) and birch (*Betula* L. spp.), with a lesser presence of other deciduous species. The estate has been used as a test site by the Swedish University of Agricultural Science (SLU) since the mid-1980s. Eahagen is a nature reserve neighboring the estate that provided additional broadleaved tree species; among these, the species considered for this study were pedunculate oak (*Quercus robur* L.) and birch. Both the estate and the reserve have forest stands with rather high homogeneity in terms of species composition (single dominant species ≥80% in volume).

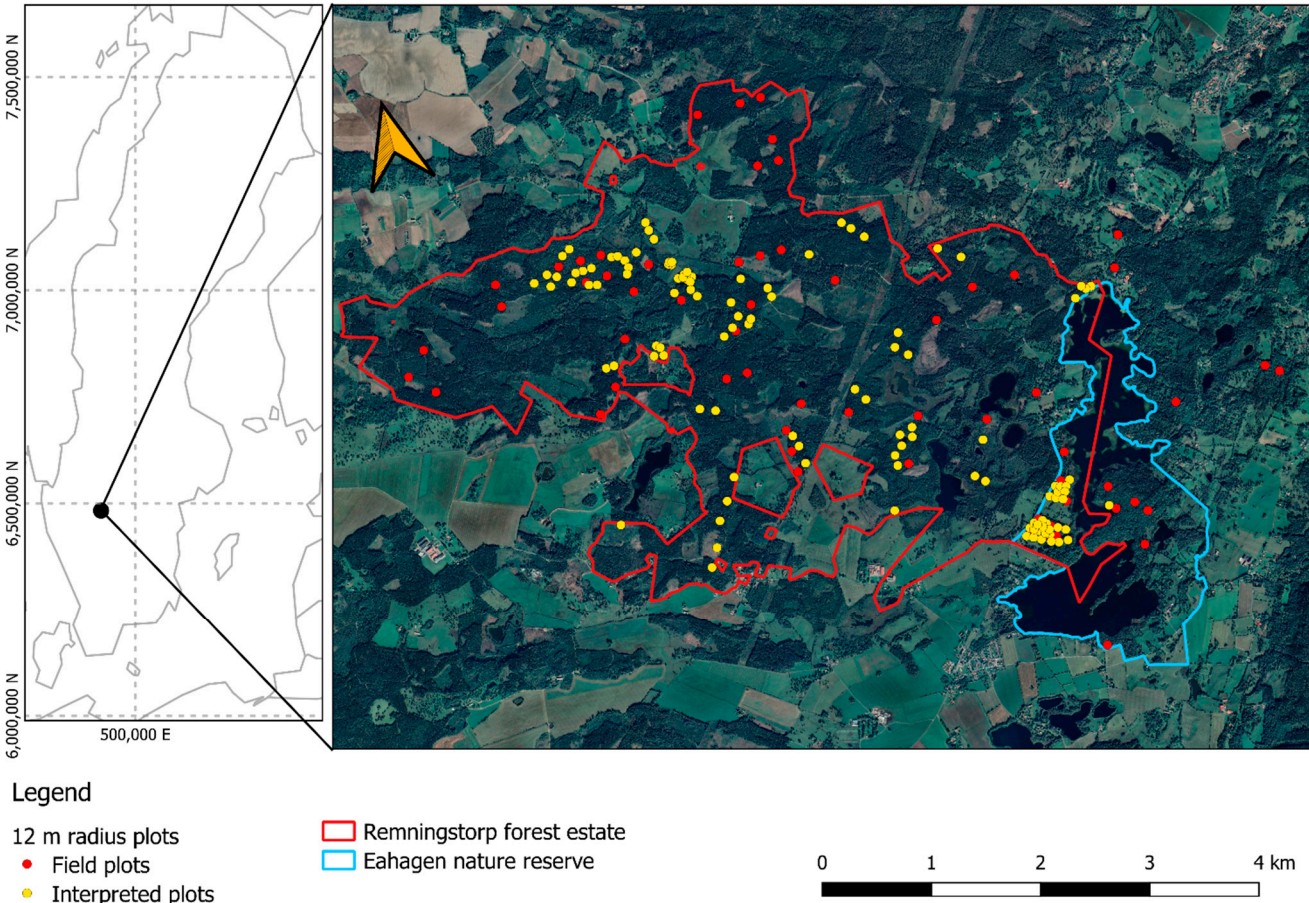

**Figure 1.** Study site location and plot distribution. The field plots in red show where measurements and information were collected on the ground; the photo-interpreted plots are yellow and were subjectively placed to extend the dataset, especially of minor classes.

### 2.2. Field Data

The field plots used for evaluation consisted of two field inventories where the first was carried out in Remningstorp between June and August of 2016. The sampling design was systematic sampling with a random starting point, with circular field plots of 12 m radius, distributed in a grid. This inventory mainly consisted of Norway spruce, Scots pine and birch stands. For the second inventory, carried out in the adjacent Eahagen nature reserve during the same period, the location of the plots' center was flexible, to find plots dominated by a single tree species within homogenous stands [31,32]. The overall number of field surveyed plots was 105 and the 62 plots with at least 80% of growing stock

belonging to a single species were selected for this study [33]. The training data consisted of 115 visually interpreted plots from aerial photos, acquired between 2017 and 2018, on the same area using the forest management plan to extract ancillary data (species, growing stock) aiming for similar proportions of each species [34,35]. The virtual 12 m plots were located in areas of the stands dominated by a single tree species, with plot centers at least 30 m apart.

All the selected plots contained more than 100 m$^3$/ha in volume, which is above the saturation level where SAR backscatter correlates with the stem volume at C-band [17,36–38]. The growing stock distribution for the field plots is illustrated in Figure 2. The target species are listed in Table 1 and regrouped into forest type (FTY) and into species (SPP) categories for the two classifications. The distribution of the plots is shown in Figure 1. A total of 177 plots were used with 62 field plots and 115 interpreted plots.

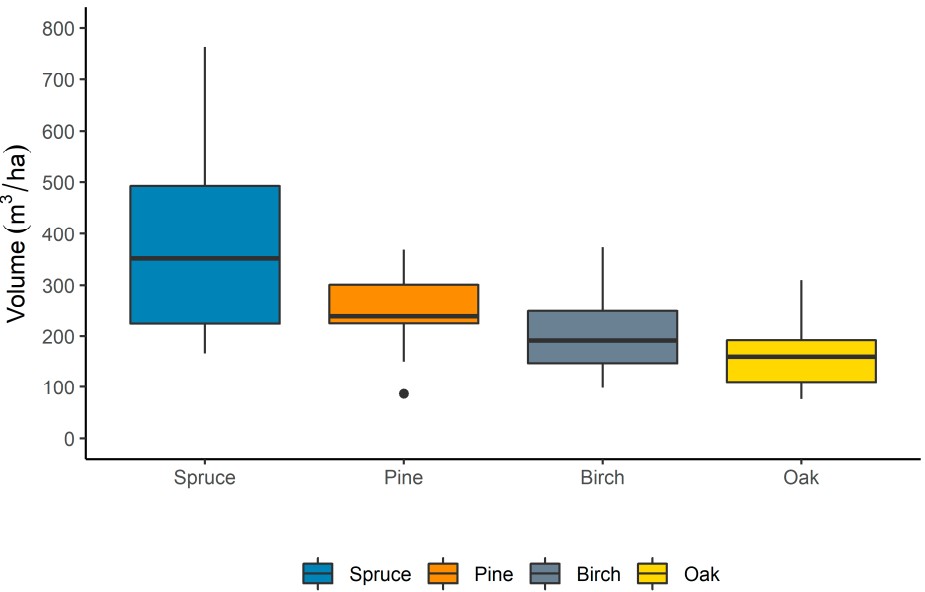

**Figure 2.** Growing stock distribution derived from the field plots in the different forest stand types (pure stands).

**Table 1.** Dominant tree species distribution in the study plots.

| Tree Species | Scientific Name | FTY | SPP | Field Plots | Interpreted Plots |
|---|---|---|---|---|---|
| Norway spruce | *Picea abies* (L.) Karst. | Coniferous | Spruce | 28 | 14 |
| Scots pine | *Pinus sylvestris* L. | Coniferous | Pine | 13 | 38 |
| Birch | *Betula* L. spp. | Deciduous | Birch | 8 | 27 |
| Oak | *Quercus robur* L. | Deciduous | Oak | 13 | 36 |

### 2.3. Satellite Data

Sentinel-1 acquisitions from January 2017 to December 2019 were downloaded from the Copernicus Open Access Hub and the Alaska Satellite Facility. The images were acquired from the same satellite orbit (73) in the interferometric wide swath mode and provided as ground range detected data in 10 m resolution and mean incidence angle of 39°. The multi-seasonal time series was selected to use the temporal signatures [28] and understand the temporal and phenological behavior of backscatter [39,40]. A total of 180 images were downloaded in both VH and VV polarizations. The number of images included for each year is reported in Table 2.

**Table 2.** Number of acquisitions from 2017 to 2019 downloaded and used as input for the pre-processing, divided per year.

| Year | No. of Acquisitions |
|------|---------------------|
| 2017 | 61 |
| 2018 | 61 |
| 2019 | 58 |
| Total | 180 |

### 2.4. Pre-Processing of Satellite Images

The satellite images were processed with the Sentinel Application Platform (SNAP) from ESA [41]. First, the SAR acquisitions were radiometrically calibrated to $\sigma^o$ values, subset to the study area, and the terrain corrected using an external laser terrain model. To reduce the influence of changing weather and environmental conditions (e.g., temperature or precipitation) that may have occurred during the acquisitions [42], the images underwent histogram matching using a summer image as reference (21 July 2018). The images were filtered using a Sigma Lee filter with a $5 \times 5$ moving window, which is effective in noise removal and enough to preserve the image spatial resolution avoiding blurring in edges and point targets. A sigma value of 0.9 was used, indicated in the literature as suitable for general applications [43–45]. As a last step, temperature correction was applied to the images. The radar backscatter is sensitive to variations in the environmental conditions, especially those driven by temperature [42]. To estimate the influence of temperature, a linear regression model (with parameters $\alpha_0, \alpha_1, \alpha_2, \alpha_3, \alpha_4$) was used to predict the backscatter values for forest pixels in the images (represented by the mean backscatter of all forest plots). Daily mean temperatures were provided by the Swedish Meteorological and Hydrological Institute for the station near the study area. The moving averages of 3, 5 and 7 days ($t_3$, $t_5$ and $t_7$, respectively) were computed and used in addition to the daily means to estimate the average backscatter level $\hat{\sigma}_{avg}$ of the scene (1). The estimated average backscatter could then be subtracted from each image $i$, with the remaining signal, the normalized backscatter ($\hat{\sigma}^o_{norm}$), being correlated to the tree species. The mean plot values were then extracted for all images and polarizations (Figure 3).

$$\hat{\sigma}^o_{avg,i} \sim \alpha_0 + \alpha_1 t_{1,i} + \alpha_2 t_{3,i} + \alpha_3 t_{5,i} + \alpha_4 t_{7,i} \tag{1}$$

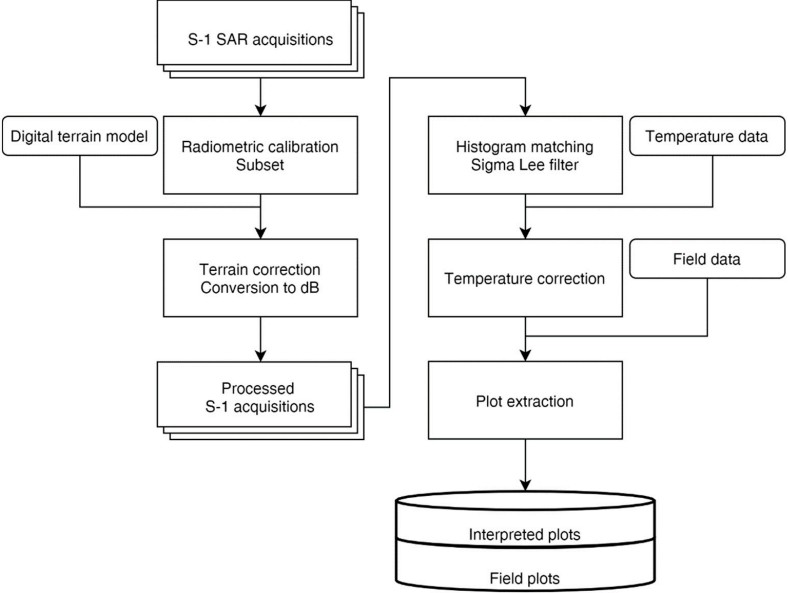

**Figure 3.** Pre-processing of the S-1 SAR data.

### 2.5. Classification and Validation

The mean plot values were used with the interpreted plot references as training data to create two Random Forest (RF) models, one to classify FTY and one for SPP [12,46]. The models were used to predict and assess the field plots (used as test data). We used the R package *randomForest* to perform the classifications with the split decision parameters *mtry* = 13 when using only one polarization, and *mtry* = 19 when combined, and a forest size of 500 trees (*ntree* = 500). All other parameters were used with the default values.

The predictions were validated with the field references and expressed as a confusion matrix with user's accuracy (UA), producer's accuracy (PA) and overall accuracy (OA) [34]. The UA describes the proportion of correctly classified sample plots, essentially as how often the class on the map will be present in the field. The PA describes the proportion of field plots that were correctly classified and, hence, describes how often real features on the ground are correctly shown on the classified map. Finally, the overall accuracy is the proportion of the data correctly classified over the total number of observations. The Cohen's kappa (*K*) coefficient was computed to give a robust measure of agreement calculation for qualitative variables to better account for reliability and validity of the classification [26,28,29,47,48]. Cohen's *K* considers both inter-rater and intra-rater reliability among the observations and observer [47]. Landis and Koch [49] proposed the following guidelines to interpret *K* coefficient: value < 0 indicates no agreement, 0–0.20 as slight, 0.21–0.40 as fair, 0.41–0.60 as moderate, 0.61–0.80 as substantial, and 0.81–1 as almost perfect agreement.

Since the satellite images are covering the same area for three years, the mean plot values across different images were assumed to be correlated. We therefore applied a Principal Component Analysis (PCA) to reduce the dimensionality and investigate the impact of using multiple images. The PCA was performed in R using the *prcomp* function of the *stats* package [50–52] on the VH and VV polarization separately, and then in combination. The first two principal components were used as input to another RF classification, performed as described above.

### 2.6. Seasonality

Since the differences are greater in the winter season [28,29], to investigate the temporal stability we created RF models for FTY based only on winter images from one ($w_2$ or $w_3$), two ($w_2$ and $w_3$), and three seasons ($w_2$, $w_3$ and $w_1$) (Table 3). The last time span ($w_4$) was not considered due to the lower number of images. The subsets comprised winter images ranging from 1 November to 1 April, adapting the approach used by Rüetschi et al. [28]. Each FTY classification was validated as described in the previous section.

**Table 3.** Winter season subsets and number of images. Only $w_2$ and $w_3$ are considered full winter seasons.

| Subset | First Date | Last Date | Number of Images |
|--------|-----------|-----------|------------------|
| $w_1$ | 1 January 2017 | 1 April 2017 | 15 |
| $w_2$ | 1 November 2017 | 1 April 2018 | 25 |
| $w_3$ | 1 November 2018 | 1 April 2019 | 23 |
| $w_4$ | 1 November 2019 | 31 December 2019 | 6 |

## 3. Results

### 3.1. Backscatter Trend

Figure 4 shows the temporal signature for deciduous and coniferous forest types, for the VH-polarization (Figure 4a) and VV-polarization (Figure 4b). Comparing the $\hat{\sigma}^o_{norm}$ values across time, similar trends were evident for both polarizations over the entire period. The $\hat{\sigma}^o_{norm}$ values were higher for deciduous compared to coniferous forest, with the biggest difference in the VH-polarization. The coniferous and deciduous forests had an opposite seasonal trend resulting in a separation of the two temporal signatures during winter. For the tree species, to better understand the trend of the signal, a moving average of

window size 5 for each species is shown in Figure 5. The $\hat{\sigma}^o_{norm}$ values over time showed the same trend observed for Figure 4, with the biggest difference among signals in the VH-polarization (Figure 5a).

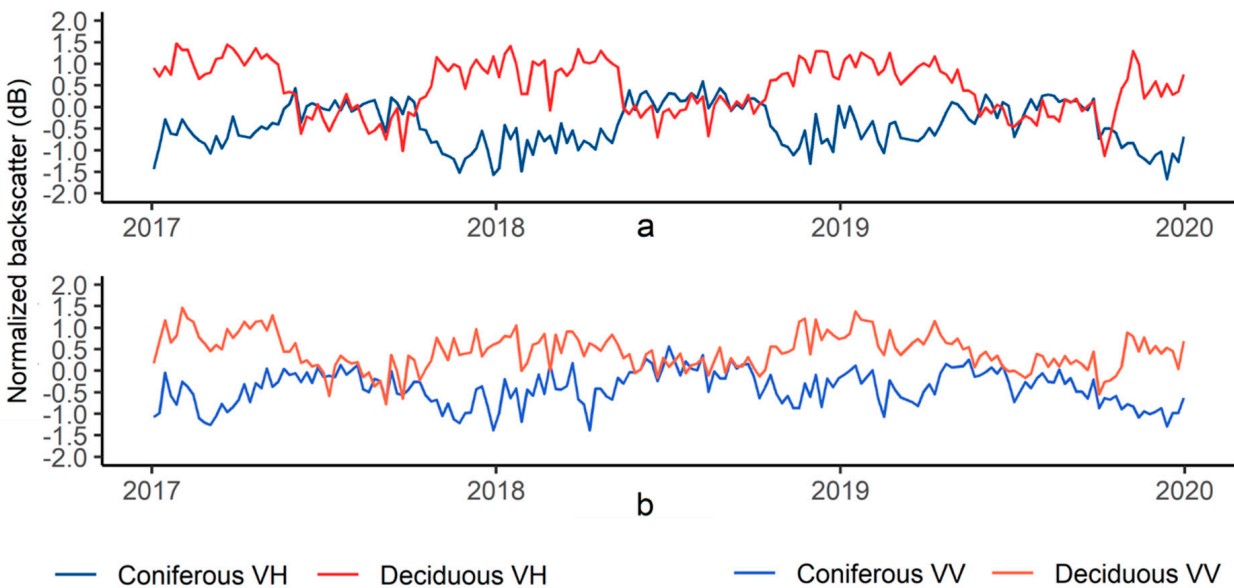

**Figure 4.** Backscatter trend over time. A comparison between Coniferous and Deciduous FTY for (**a**) VH-polarization and (**b**) VV-polarization.

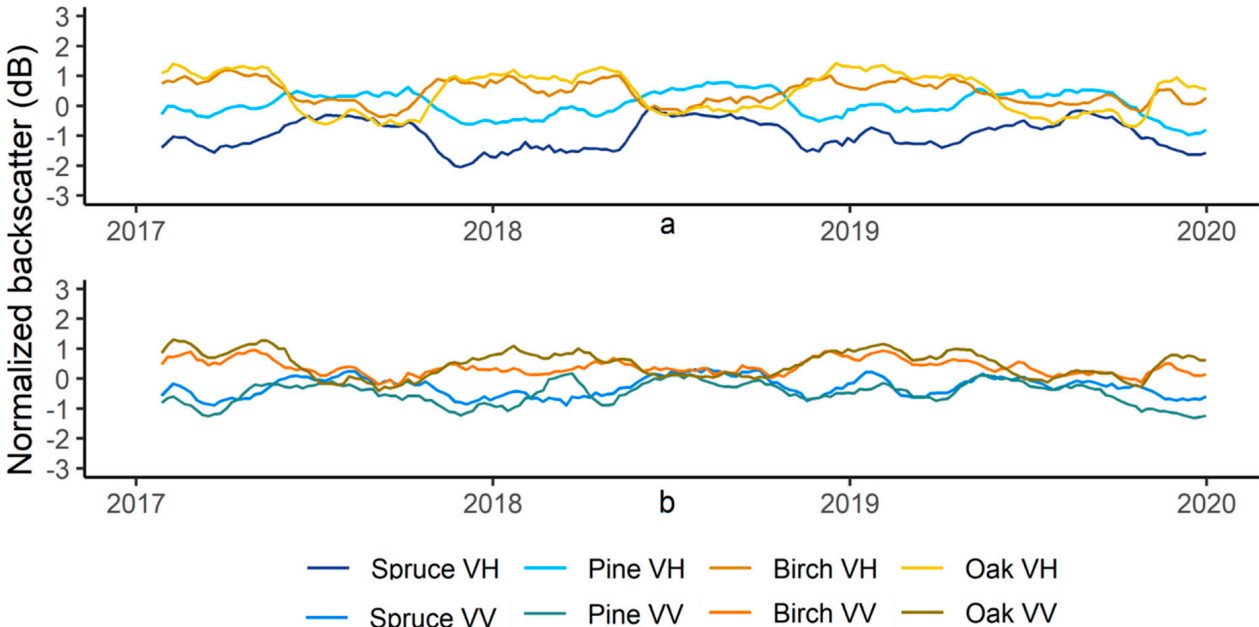

**Figure 5.** Backscatter trend over time. A comparison between moving averages (window size 5) for the tree species investigated for (**a**) VH-polarization and (**b**) VV-polarization.

### 3.2. Forest Type Classification

The overall accuracy results for the RF classification models for forest type (FTY), according to the polarization used, are presented in Table 4. The use of only VH-polarization images resulted in higher OA, compared to using VV or VH + VV [29]. The use of dual-polarizations, also present in the literature [26,28,53], accomplished good results, especially

after applying the PCA. In general, the OA improved after performing the PCA, with the exception of FTY predicted with only VH-pol data. The backscatter was prominently separated on average, although having a certain overlap at the plot level (Figure 6). The separation was similar for both polarizations, with the VH-polarization covering a larger range. The classification of FTY using the VH + VV combination after PCA is illustrated as a map over the study area in Figure 7.

Table 5 presents the results of the FTY classification before PCA using the VH + VV combination as input, and Table 6 presents the results after PCA The first two columns in both tables list the number of plots used for the validation of the RF models.

**Table 4.** Overall accuracy values for forest type (FTY) classifications, according to the polarization used, before and after the PCA.

| Polarization | OA before PCA | OA after PCA |
|:---:|:---:|:---:|
| VH | 0.94 | 0.92 |
| VV | 0.73 | 0.84 |
| VH + VV | 0.89 | 0.94 |

**Table 5.** FTY classification results using VH + VV polarization data. Producer's accuracy (PA), user's accuracy (UA), overall accuracy (OA) and Cohen's kappa coefficient (*K*) are also reported.

| Confusion Matrix | | | | | |
|:---:|:---:|:---:|:---:|:---:|:---:|
| **Reference** | **Classification** | | **PA** | **OA** | *K* |
| | **Coniferous** | **Deciduous** | | | |
| **Coniferous** | 36 | 5 | 0.88 | 0.89 | 0.75 |
| **Deciduous** | 2 | 19 | 0.90 | | |
| **UA** | 0.95 | 0.78 | | | |

**Table 6.** FTY classification results after PCA derived from VH + VV polarization data. PA, UA, OA and *K* are also reported.

| Confusion Matrix | | | | | |
|:---:|:---:|:---:|:---:|:---:|:---:|
| **Reference** | **Classification** | | **PA** | **OA** | *K* |
| | **Coniferous** | **Deciduous** | | | |
| **Coniferous** | 37 | 4 | 0.90 | 0.94 | 0.86 |
| **Deciduous** | 0 | 21 | 1.00 | | |
| **UA** | 1.00 | 0.84 | | | |

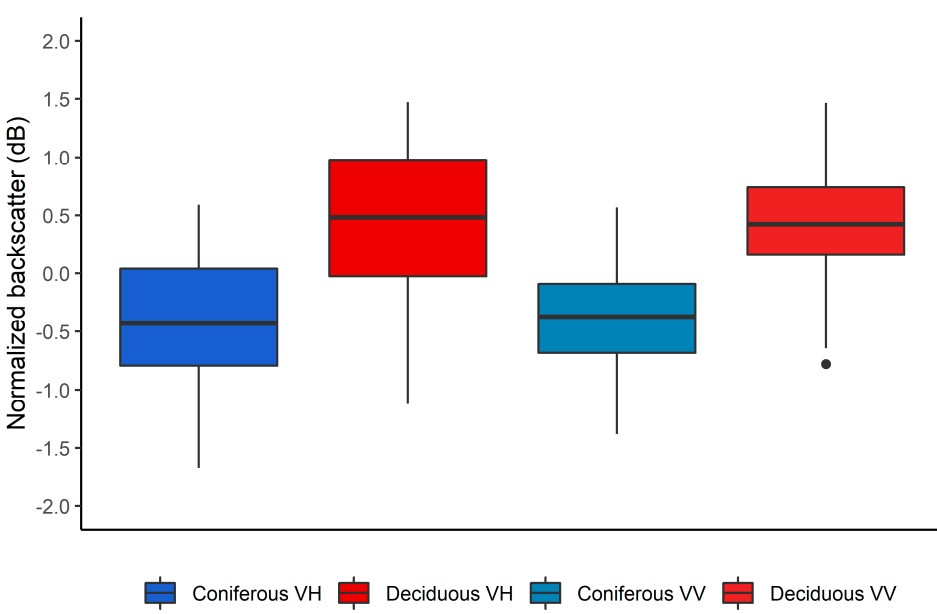

**Figure 6.** Normalized backscatter distribution with reference to forest type (FTY) and polarization.

Coniferous VH ■ Deciduous VH ■ Coniferous VV ■ Deciduous VV

■ Coniferous
■ Deciduous
□ Remningstorp forest estate
□ Eahagen nature reserve

**Figure 7.** Classification for forest type (FTY) over the study area and the surrounding forests as reported in Table 6.

The accuracy (PA, UA, OA and *K*) improved after PCA, with *K* increasing from 0.75 to 0.86. The *K* indicated a substantial and almost perfect agreement between the observed

and classified values, before and after PCA, respectively. Both classifications, before and after PCA, achieved high values for PA and UA for the coniferous FTY, equal to or higher than 0.88 (Tables 5 and 6). For the deciduous FTY, the accuracy results were lower but still encouraging and higher than 0.78 (Tables 5 and 6).

### 3.3. Species Classification

Compared to FTY, the SPP classification achieved lower values in accuracy. The overall accuracy results for the RF classification models for tree species (SPP), according to the polarization used, are presented in Table 7. The highest OA was obtained when using the VH-polarization only, compared to VV and VH + VV. After applying the PCA, the accuracies improved for all polarizations. The classification of SPP using the VH data after PCA is illustrated as a map over the study area in Figure 8.

**Table 7.** Overall accuracy values for tree species (SPP) classifications, according to the polarization used, before and after the PCA.

| Polarization | OA before PCA | OA after PCA |
|:---:|:---:|:---:|
| VH | 0.55 | 0.66 |
| VV | 0.35 | 0.48 |
| VH + VV | 0.50 | 0.53 |

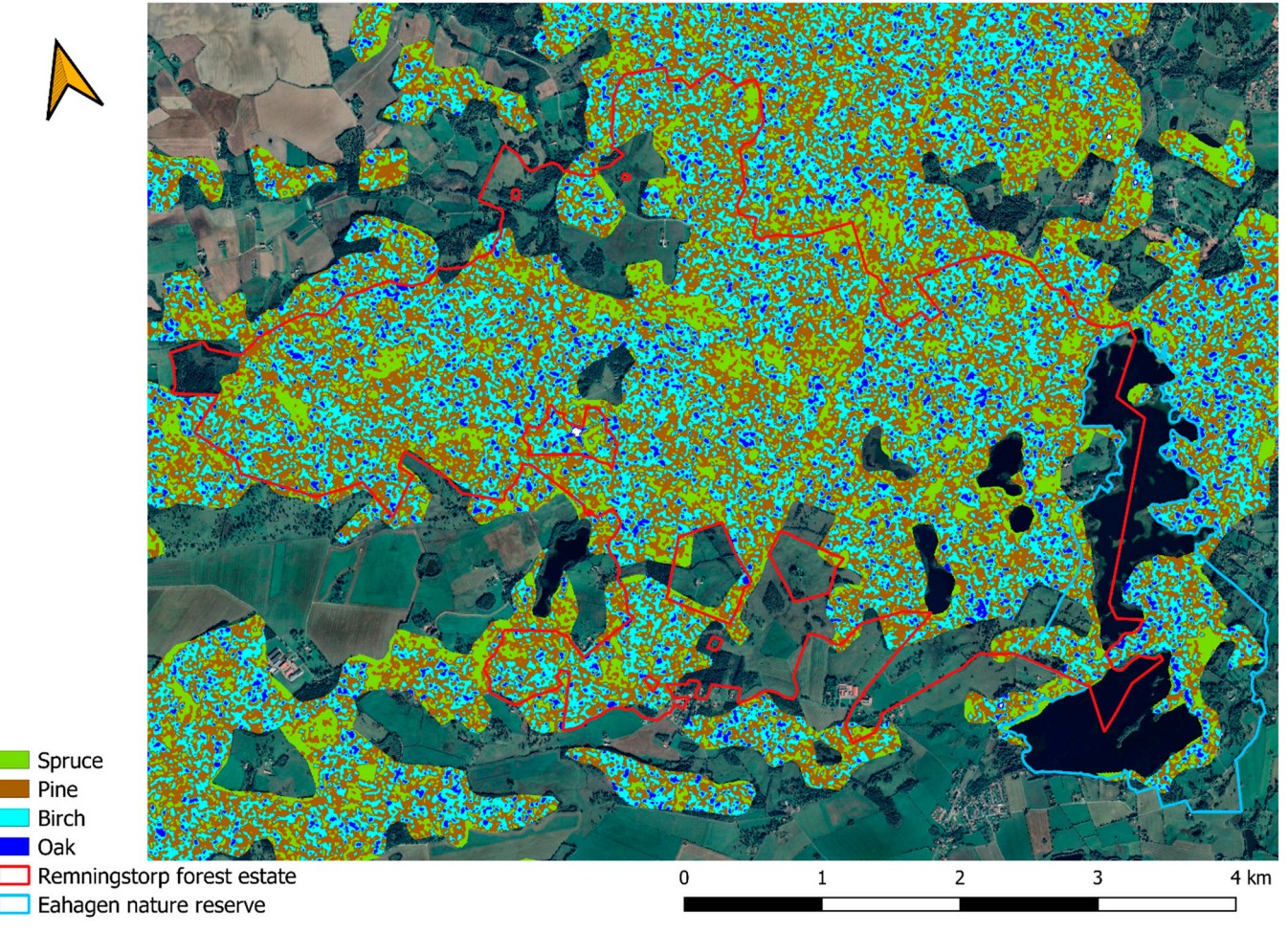

**Figure 8.** Classification for species classification (SPP) over the study area and the surrounding forests as reported in Table 9.

The confusion matrices for SPP classification using VH data before and after PCA are presented in Tables 8 and 9. The accuracies were slightly lower compared to the FTY

classification, mainly caused by misclassification between spruce and pine. A similar ambiguity could be seen between oak and birch.

**Table 8.** SPP classification results using VH polarization. PA, UA, OA and *K* are also reported.

| | Confusion Matrix | | | | | | |
|---|---|---|---|---|---|---|---|
| | **Classification** | | | | **PA** | **OA** | **K** |
| **Reference** | **Birch** | **Oak** | **Pine** | **Spruce** | | | |
| **Birch** | 1 | 7 | 0 | 0 | 0.13 | | |
| **Oak** | 0 | 11 | 1 | 1 | 0.85 | | |
| **Pine** | 2 | 0 | 11 | 0 | 0.85 | 0.55 | 0.39 |
| **Spruce** | 0 | 1 | 16 | 11 | 0.39 | | |
| **UA** | 0.33 | 0.58 | 0.39 | 0.92 | | | |

**Table 9.** SPP classification results from principal components derived from VH polarization data. PA, UA, OA and *K* are also reported.

| | Confusion Matrix | | | | | | |
|---|---|---|---|---|---|---|---|
| | **Classification** | | | | **PA** | **OA** | **K** |
| **Reference** | **Birch** | **Oak** | **Pine** | **Spruce** | | | |
| **Birch** | 3 | 5 | 0 | 0 | 0.38 | | |
| **Oak** | 1 | 12 | 0 | 0 | 0.92 | | |
| **Pine** | 3 | 0 | 9 | 1 | 0.69 | 0.66 | 0.54 |
| **Spruce** | 0 | 2 | 9 | 17 | 0.61 | | |
| **UA** | 0.43 | 0.63 | 0.50 | 0.94 | | | |

Despite overall higher values in accuracy (UA, PA, OA and *K*), misclassified spruce plots had an impact not only on pine, but also on the deciduous species. For both Cohens' *K* values reported in Tables 8 and 9, the degree of agreement between observed and classified values was generally classified as fair and moderate. Among the SPP classifications, the VV-polarization performed poorly, but also the VH + VV combination did not surpass the VH-polarization either in OA (0.53), or the Cohen's coefficient (*K* = 0.39).

### 3.4. Seasonality

Overall, the use of the whole set of images available (three full years) achieved a high level of accuracy. We investigated the temporal stability creating RF models for classifying the FTY based on single or multiple sets of winter images. The OA values are reported in Table 10. In general, the models did not achieve the same level of accuracy as reported in Table 5 for FTY. The use of multiple seasons helped to increase the accuracy for the VH and VV polarization compared to the use of a single season, whereas no major improvement was noted for the VH + VV combination. After performing the PCA, the accuracy values did not improve as observed before.

Confusion matrices and additional results from all RF combinations (with respect to polarizations, seasons, FTY and SPP) are reported in Appendix A.

**Table 10.** Overall accuracy for FTY classification using images from winter seasons.

| | OA before PCA | | | OA after PCA | | |
|---|---|---|---|---|---|---|
| | **VH** | **VV** | **VH + VV** | **VH** | **VV** | **VH + VV** |
| 1 winter season ($w_2$ or $w_3$) | 0.85 | 0.73 | 0.87 | 0.82 | 0.69 | 0.83 |
| 2 winter seasons ($w_2 + w_3$) | 0.89 | 0.76 | 0.85 | 0.82 | 0.65 | 0.84 |
| 3 winter seasons ($w_2 + w_3 + w_1$) | 0.89 | 0.79 | 0.87 | 0.81 | 0.71 | 0.85 |

## 4. Discussion

The evaluation of accuracies showed that the adopted approach is suitable for classifying FTY and SPP. The highest OA result was obtained from merely the VH-polarization, confirming that this polarization contained most of the information related to tree species [29], however, the models based on dual-polarization also performed well. We expected that the use of both polarizations and the whole three-year period would provide the most information to achieve the best classification result, but the OA for the VH + VV combination reported in Table 4 was lower than the OA obtained from VH-polarization only. In general, comparing the classification results before and after PCA (Table 4), an improvement in the accuracy values can be observed for both FTY and SPP, which may be connected to the reduction in the input dimensions, but still managing not to overfit the models [46].

The classification of FTY showed that our results are comparable to several past studies, using fully polarimetric C-band SAR data [25,40,54,55]. Compared to more recent dual-pol studies with similar approaches, both the OA and Cohen's coefficient matched the results in these works, where the OA ranged between 0.77 and 0.86 and $K = 0.42$–0.73 [28,29]. A more detailed comparison is reported in Table 11. One of the study sites analyzed by Dostálová et al. [29] was the same as that used in this study, but their results showed a lower accuracy, mainly due to poor reference forest type data. When SAR have been combined with optical data [55], accuracies have been reported in the same range as the ones we obtained.

**Table 11.** Results comparison with previous studies in the literature.

| | Site Location | Coniferous | | Deciduous | | OA | K |
|---|---|---|---|---|---|---|---|
| | | UA | PA | UA | PA | | |
| Rüetschi et al. [28] | Switzerland (CH) | 0.84 | 0.88 | 0.88 | 0.84 | 0.86 | 0.73 |
| Dostálová et al. [29] | Neusiedl Lake (AT) | 0.46 | 0.73 | 0.77 | 0.69 | 0.85 | 0.69 |
| | Remningstorp (SE) | 0.72 | 0.67 | 0.38 | 0.42 | 0.77 | 0.60 |
| | Krycklan (SE) | 0.69 | 0.69 | 0.19 | 0.21 | 0.65 | 0.42 |
| Bjerreskov et al. [55] * | Denmark (DK) | 0.95 | 0.95 | 0.96 | 0.96 | 0.95 | - |
| Present study (Table 6) | Remningstorp (SE) | 1.00 | 0.90 | 0.84 | 1.00 | 0.94 | 0.86 |

\* input data are from the fusion of multi-temporal Sentinel-1 and 2 data.

As expected from previous works in the literature [26,28,29,55], the SPP classification did not reach the same level of accuracy as the FTY classification. There are not many studies concerning species classification, but our results (Table 8) are in line with previous studies, where the OA ranged from 0.63 [55] to 0.72 [28]. The coniferous tree species showed a bigger difference in normalized backscatter between each other than the deciduous (Figure 5a,b).

The use of multiple seasons and multiple years of acquisitions influenced the predicted accuracy to varying degrees. Our efforts concentrated on the use of single and multiple winter periods in order to take advantage of the leaf-off and leaf-on conditions. The use of PCA on the winter images did not improve the accuracy or Cohen's coefficient. Yet, the use of multiple winter seasons reached, on average, higher results in accuracy than with the use of only a single season. Still, the use of VH-polarization alone reached the same results in average OA, compared to the use of dual-polarization. The Cohen's coefficient changed similarly to OA, but with values in the range of $K = 0.43$–0.76 without the use of PCA, and $K = 0.32$–0.68 when RF was performed after PCA (Figures A1 and A2).

Compared to previous works, the difference in the results was large and may have been the outcome of the application of different approaches and models. We used the mean plot value compared to previous studies, where the single pixel was selected as spatial unit [26,28,29,55]. Some studies that combined C-band SAR with additional data sources were able to reach higher classification accuracy [28,55], showing that the use of data from different sources (i.e., laser, optical, etc.) and multitemporal data involving SAR is promising, although further exploration is needed.

The radar backscatter is influenced by the acquisition configuration, and to reduce unintended effects from different configurations, we used the same incidence angle, orbit and polarization for all images. The temperature was the only weather factor considered and for which the data was corrected. The relationship with the radar backscatter when classifying forests has been investigated previously [28]; e.g., for their deciduous species (oak and beech) the backscatter was increasing with the decrease in temperature, and spruce's backscatter was decreasing with the decrease in temperature, showing that the two different forest types react differently when the temperature varies. There is also a factor related to interspecific and intraspecific variability to consider. Different trees have different phenology and different geometric characteristics, even within the same species or the same population. For example, the timing of leaf development and start of the growing season varies within single species.

A final consideration was related to the backscatter saturation threshold of biomass. Figure 2 shows the growing stock distribution for the field plots in the different forest stand types. The C-band SAR backscatter reaches the saturation point of biomass earlier compared to longer wavelengths [17,36,38]. The growing stock for each tree species surpassed the established saturation threshold of 100 $m^3$/ha, making the signal less sensitive to volume variation. For the final classification and accuracy assessment, we assumed the volume saturation did not therefore influence the results.

## 5. Conclusions

The paper evaluated the accuracy for forest type and tree species classification using C-band SAR backscatter. We also assessed the influence of using multiple seasons and multiple acquisition years on the predicted accuracy for the forest type classification. The main findings were:

i.      The proposed approach showed good results with the FTY classification overall accuracy reaching 94%. We obtained high values for both producer's and user's accuracy, ranging between 84 and 100%, which was convincing also compared to similar studies [28,29,55]. Moreover, the RF model for the classification achieved high values also in Cohen's K, indicating a high degree of agreement between field and predicted values. The accuracy results indicate that this method is suitable for the creation and use of FTY maps (Figure 6) [12,56].

ii.     Compared to the FTY classification, the results from the SPP classification showed more errors compared to the field values, with a maximum OA of 66%. This result was similar to comparable studies [28,55].

iii.    The use of multiple winter seasons delivered better accuracies compared to the use of single winter seasons. The VH polarization contained most of the information and by using the VH + VV combination, the results improved slightly. The differences between forest types were biggest during the winters and by using winter images the results were almost as high as using all year round images.

iv.     The use of PCA generally improved the classifications of both forest type and tree species, although this was not the case when using only winter images.

The classification of FTYs showed better results compared to the classification of tree species, and to improve the accuracies of the SPP classification, the most promising approach would be to combine SAR with other RS sensors [53,55,57,58]. Additionally, the combination of radar data with different wavelengths or higher resolution could possibly provide more information.

We used plots with only pure species in this study (with at least 80% of the volume). The forest landscape was not composed only of pure stands but of clusters of trees of the same species. Future research should address this topic, aiming at successful classifications (both FTY and SPP) also in mixed stands, i.e., where there is not a dominant species. We successfully assessed the use of C-band SAR data to perform forest type and tree species classification, although there is still room for improving the classification accuracy by experimenting new classification algorithms and new techniques.

**Author Contributions:** Conceptualization, A.U. and H.J.P.; methodology, A.U. and H.J.P.; software and validation, A.U.; formal analysis, E.L.; investigation and data curation, A.U. and H.J.P.; writing—original draft preparation, A.U.; writing—review and editing, A.U., H.J.P. and E.L.; visualization, A.U., H.J.P. and E.L.; supervision, H.J.P. All authors have read and agreed to the published version of the manuscript.

**Funding:** The research was funded by FORMAS (2018-01161_3), Skogssällskapet (Nya skogskartor med trädslagsinformation från tidsserier av satellitdata, 2019-660), and by the Hildur & Sven Wingquist's Foundation for Forest Research.

**Institutional Review Board Statement:** Not applicable.

**Informed Consent Statement:** Not applicable.

**Data Availability Statement:** The data presented in this study are available on request from the corresponding author.

**Conflicts of Interest:** The authors declare no conflict of interest.

### Appendix A

RF models and classifications' results are here reported for further understanding of the main text results.

**Table A1.** FTY classification results using VH polarization data. Accuracies metrics (PA, UA, OA and *K*) are also reported.

| | Confusion Matrix | | | | |
|---|---|---|---|---|---|
| Reference | Classification | | PA | OA | *K* |
| | Coniferous | Deciduous | | | |
| Coniferous | 38 | 3 | 0.93 | | |
| Deciduous | 1 | 20 | 0.95 | 0.94 | 0.86 |
| UA | 0.97 | 0.87 | | | |

**Table A2.** FTY classification results using VV polarization data. Accuracies metrics (PA, UA, OA and *K*) are also reported.

| | Confusion Matrix | | | | |
|---|---|---|---|---|---|
| Reference | Classification | | PA | OA | *K* |
| | Coniferous | Deciduous | | | |
| Coniferous | 27 | 14 | 0.66 | | |
| Deciduous | 3 | 18 | 0.86 | 0.73 | 0.46 |
| UA | 0.90 | 0.56 | | | |

**Table A3.** FTY classification results using the VH + VV combination. Accuracies metrics (PA, UA, OA and *K*) are also reported.

| | Confusion Matrix | | | | |
|---|---|---|---|---|---|
| Reference | Classification | | PA | OA | *K* |
| | Coniferous | Deciduous | | | |
| Coniferous | 36 | 5 | 0.88 | | |
| Deciduous | 2 | 19 | 0.90 | 0.89 | 0.76 |
| UA | 0.95 | 0.79 | | | |

**Table A4.** FTY classification results from PCA derived from VH polarization data. Accuracies metrics (PA, UA, OA and *K*) are also reported.

| Confusion Matrix | | | | | |
| --- | --- | --- | --- | --- | --- |
| Reference | Classification | | PA | OA | *K* |
| | Coniferous | Deciduous | | | |
| Coniferous | 36 | 5 | 0.88 | 0.92 | 0.83 |
| Deciduous | 0 | 21 | 1.00 | | |
| UA | 1.00 | 0.81 | | | |

**Table A5.** FTY classification results from PCA derived from VV polarization data. Accuracies metrics (PA, UA, OA and *K*) are also reported.

| Confusion Matrix | | | | | |
| --- | --- | --- | --- | --- | --- |
| Reference | Classification | | PA | OA | *K* |
| | Coniferous | Deciduous | | | |
| Coniferous | 32 | 9 | 0.78 | 0.84 | 0.67 |
| Deciduous | 1 | 20 | 0.95 | | |
| UA | 0.97 | 0.69 | | | |

**Table A6.** FTY classification results from PCA derived from the VH + VV combination. Accuracies metrics (PA, UA, OA and *K*) are also reported.

| Confusion Matrix | | | | | |
| --- | --- | --- | --- | --- | --- |
| Reference | Classification | | PA | OA | *K* |
| | Coniferous | Deciduous | | | |
| Coniferous | 37 | 4 | 0.90 | 0.94 | 0.86 |
| Deciduous | 0 | 21 | 1.00 | | |
| UA | 1.00 | 0.84 | | | |

**Table A7.** SPP classification results using VH polarization. Accuracies metrics (PA, UA, OA and *K*) are also reported.

| Confusion Matrix | | | | | | | |
| --- | --- | --- | --- | --- | --- | --- | --- |
| Reference | Classification | | | | PA | OA | *K* |
| | Birch | Oak | Pine | Spruce | | | |
| Birch | 1 | 7 | 0 | 0 | 0.13 | 0.55 | 0.39 |
| Oak | 0 | 11 | 1 | 1 | 0.85 | | |
| Pine | 2 | 0 | 11 | 0 | 0.85 | | |
| Spruce | 0 | 1 | 16 | 11 | 0.39 | | |
| UA | 0.33 | 0.58 | 0.39 | 0.92 | | | |

**Table A8.** SPP classification results using VV polarization. Accuracies metrics (PA, UA, OA and *K*) are also reported.

| Reference | Confusion Matrix | | | | PA | OA | K |
|---|---|---|---|---|---|---|---|
| | Classification | | | | | | |
| | Birch | Oak | Pine | Spruce | | | |
| Birch | 2 | 5 | 1 | 0 | 0.25 | | |
| Oak | 1 | 9 | 3 | 0 | 0.69 | 0.35 | 0.19 |
| Pine | 0 | 2 | 11 | 0 | 0.85 | | |
| Spruce | 2 | 4 | 22 | 0 | 0.00 | | |
| UA | 0.40 | 0.45 | 0.30 | - | | | |

**Table A9.** SPP classification results using the VH + VV combination. Accuracies metrics (PA, UA, OA and *K*) are also reported.

| Reference | Confusion Matrix | | | | PA | OA | K |
|---|---|---|---|---|---|---|---|
| | Classification | | | | | | |
| | Birch | Oak | Pine | Spruce | | | |
| Birch | 1 | 7 | 0 | 0 | 0.13 | | |
| Oak | 0 | 11 | 2 | 0 | 0.85 | 0.50 | 0.35 |
| Pine | 0 | 1 | 12 | 0 | 0.92 | | |
| Spruce | 0 | 2 | 19 | 7 | 0.25 | | |
| UA | 1.00 | 0.52 | 0.36 | 1.00 | | | |

**Table A10.** SPP classification results from PCA derived from VH polarization data. Accuracies metrics (PA, UA, OA and *K*) are also reported.

| Reference | Confusion Matrix | | | | PA | OA | K |
|---|---|---|---|---|---|---|---|
| | Classification | | | | | | |
| | Birch | Oak | Pine | Spruce | | | |
| Birch | 3 | 5 | 0 | 0 | 0.38 | | |
| Oak | 1 | 12 | 0 | 0 | 0.92 | 0.66 | 0.54 |
| Pine | 3 | 0 | 9 | 1 | 0.69 | | |
| Spruce | 0 | 2 | 9 | 17 | 0.61 | | |
| UA | 0.43 | 0.63 | 0.50 | 0.94 | | | |

**Table A11.** SPP classification results from PCA derived from VV polarization data. Accuracies metrics (PA, UA, OA and *K*) are also reported.

| Reference | Confusion Matrix | | | | PA | OA | K |
|---|---|---|---|---|---|---|---|
| | Classification | | | | | | |
| | Birch | Oak | Pine | Spruce | | | |
| Birch | 5 | 2 | 1 | 0 | 0.63 | | |
| Oak | 3 | 10 | 0 | 0 | 0.77 | 0.48 | 0.35 |
| Pine | 3 | 0 | 10 | 0 | 0.77 | | |
| Spruce | 7 | 3 | 13 | 5 | 0.18 | | |
| UA | 0.28 | 0.67 | 0.42 | 1.00 | | | |

**Table A12.** SPP classification results from PCA derived from the VH + VV combination. Accuracies metrics (PA, UA, OA and *K*) are also reported.

| Reference | Confusion Matrix | | | | PA | OA | K |
|---|---|---|---|---|---|---|---|
| | Classification | | | | | | |
| | Birch | Oak | Pine | Spruce | | | |
| **Birch** | 4 | 4 | 0 | 0 | 0.50 | | |
| **Oak** | 2 | 11 | 0 | 0 | 0.85 | 0.53 | 0.39 |
| **Pine** | 1 | 0 | 10 | 2 | 0.77 | | |
| **Spruce** | 1 | 3 | 16 | 8 | 0.29 | | |
| **UA** | 0.50 | 0.61 | 0.38 | 0.80 | | | |

**Table A13.** Seasonality: accuracy indicators of RF model performed with VH polarization data.

| | | $w_2$ | $w_3$ | $w_2 + w_3$ | $w_1 + w_2 + w_3$ |
|---|---|---|---|---|---|
| Coniferous | PA | 0.85 | 0.78 | 0.88 | 0.88 |
| | UA | 0.95 | 0.94 | 0.95 | 0.95 |
| Deciduous | PA | 0.90 | 0.90 | 0.90 | 0.90 |
| | UA | 0.76 | 0.68 | 0.79 | 0.79 |
| | OA | 0.87 | 0.82 | 0.89 | 0.89 |
| | K | 0.72 | 0.63 | 0.76 | 0.76 |

**Table A14.** Seasonality: accuracy indicators of RF model performed with VV polarization data.

| | | $w_2$ | $w_3$ | $w_2 + w_3$ | $w_1 + w_2 + w_3$ |
|---|---|---|---|---|---|
| Coniferous | PA | 0.68 | 0.63 | 0.68 | 0.73 |
| | UA | 0.90 | 0.90 | 0.93 | 0.94 |
| Deciduous | PA | 0.86 | 0.86 | 0.90 | 0.90 |
| | UA | 0.58 | 0.55 | 0.59 | 0.63 |
| | OA | 0.74 | 0.71 | 0.76 | 0.65 |
| | K | 0.48 | 0.43 | 0.52 | 0.32 |

**Table A15.** Seasonality: accuracy indicators of RF model performed using the VH + VV combination.

| | | $w_2$ | $w_3$ | $w_2 + w_3$ | $w_1 + w_2 + w_3$ |
|---|---|---|---|---|---|
| Coniferous | PA | 0.88 | 0.80 | 0.83 | 0.85 |
| | UA | 0.95 | 0.94 | 0.94 | 0.95 |
| Deciduous | PA | 0.90 | 0.90 | 0.90 | 0.90 |
| | UA | 0.79 | 0.70 | 0.73 | 0.76 |
| | OA | 0.89 | 0.84 | 0.85 | 0.87 |
| | K | 0.76 | 0.66 | 0.69 | 0.72 |

**Table A16.** Seasonality: accuracy indicators of RF model performed after PCA derived from VH polarization data.

| | | $w_2$ | $w_3$ | $w_2 + w_3$ | $w_1 + w_2 + w_3$ |
|---|---|---|---|---|---|
| Coniferous | PA | 0.78 | 0.80 | 0.80 | 0.78 |
| | UA | 0.89 | 0.94 | 0.92 | 0.91 |
| Deciduous | PA | 0.81 | 0.90 | 0.86 | 0.86 |
| | UA | 0.65 | 0.70 | 0.69 | 0.67 |
| | OA | 0.79 | 0.84 | 0.82 | 0.81 |
| | K | 0.56 | 0.66 | 0.63 | 0.60 |

**Table A17.** Seasonality: accuracy indicators of RF model performed after PCA derived from VV polarization data.

|  |  | $w_2$ | $w_3$ | $w_2 + w_3$ | $w_1 + w_2 + w_3$ |
|---|---|---|---|---|---|
| Coniferous | PA | 0.63 | 0.59 | 0.56 | 0.66 |
|  | UA | 0.87 | 0.89 | 0.85 | 0.87 |
| Deciduous | PA | 0.81 | 0.86 | 0.81 | 0.81 |
|  | UA | 0.53 | 0.51 | 0.49 | 0.55 |
|  | OA | 0.69 | 0.68 | 0.65 | 0.71 |
|  | $K$ | 0.39 | 0.38 | 0.32 | 0.42 |

**Table A18.** Seasonality: accuracy indicators of RF model performed after PCA derived from the VH + VV combination.

|  |  | $w_2$ | $w_3$ | $w_2 + w_3$ | $w_1 + w_2 + w_3$ |
|---|---|---|---|---|---|
| Coniferous | PA | 0.80 | 0.83 | 0.83 | 0.88 |
|  | UA | 0.92 | 0.92 | 0.92 | 0.90 |
| Deciduous | PA | 0.86 | 0.86 | 0.86 | 0.81 |
|  | UA | 0.69 | 0.72 | 0.72 | 0.77 |
|  | OA | 0.82 | 0.84 | 0.84 | 0.85 |
|  | $K$ | 0.63 | 0.66 | 0.66 | 0.55 |

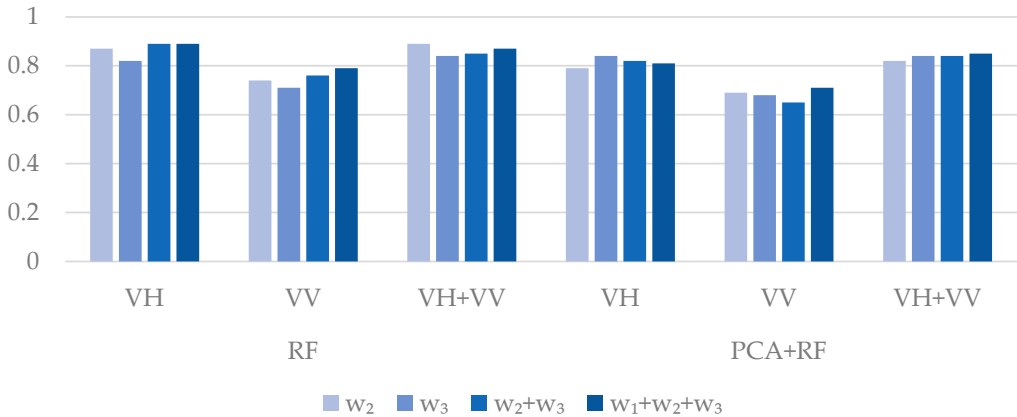

**Figure A1.** Comparison of OA indicators among the different combination of polarization (VH, VV and VH + VV) and RF model, with and without performing PCA.

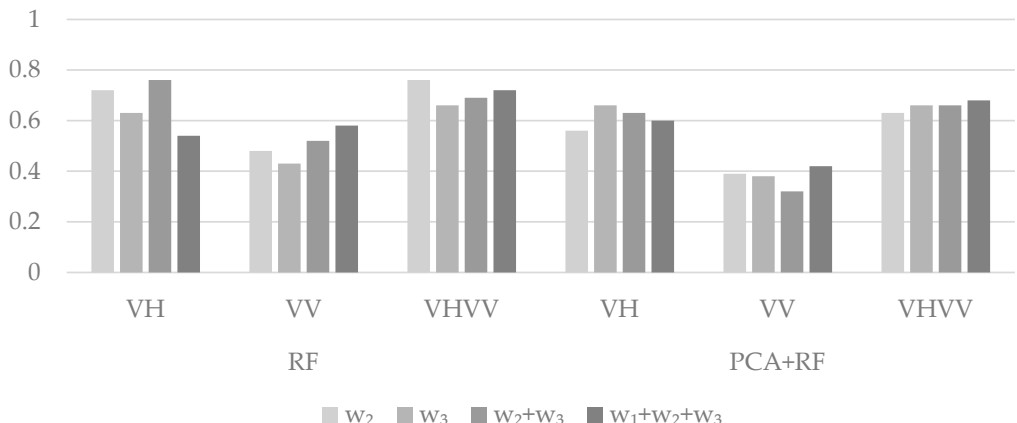

**Figure A2.** Comparison of Cohen's K among the different combination of polarization (VH, VV and VH + VV) and RF model, with and without performing PCA.

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
