# Peer review of "Assessing Forest Type and Tree Species Classification Using Sentinel-1 C-Band SAR Data in Southern Sweden"

_remotesensing, doi:10.3390/rs13163237_

Round 1

Reviewer 1 Report

Dear Authors,

I have reviewed your manuscript entitled “Assessing Forest type and tree species classification using Sentinel-1 C-band SAR data in southern Sweden”. Despite the interesting topic, it suffers from serious shortcomings.

You need to improve the introduction and methodology sections. The presented Figures and their captions don’t have acceptable quality and details. You need to provide appropriate answers to some questions focusing on random forest, time of field sampling and other ambiguous parts which are indicated in the enclosed pdf.

The manuscript needs to be revised in terms of English, some sentences are complicated and vague.

Reviewer 2 Report

Authors have tried to demonstrate potential application of Sentinel-1 timeseries data for classification of forest types and discrimination of species. Seasonality is also another factor considered for image acquisition and evaluating classification performances. The topic is interesting, but the manuscript has lack of organization, results were not presented in line with objectives. In some places results and issues discussed simultaneously, while there is a separate discussion section. Specific comments are attached. The manuscript requires major revision before accepting for publication. 

Round 2

Reviewer 1 Report

The authors have revised the manuscript based on my comments, except in one case which is very important. The classified maps of the species are required. If there are too many maps, they can show them in the supplementary section.

Author Response

Dear reviewer, 

we understood the importance of providing a map also for species classification as we did for forest type. We implemented it, together with the caption, in the text on lines 270-271.

Reviewer 2 Report

The revised version looks Okay.

Author Response

Dear reviewer,

we are pleased to see that our revision of the manuscript followed your feedback. Thank you so much for the valuable comments.